# *Ginkgo biloba*: Antioxidant Activity and In Silico Central Nervous System Potential

**Eduardo Suárez-González [1,2], Jesús Sandoval-Ramírez [3], Jorge Flores-Hernández [2,*] and Alan Carrasco-Carballo [1,*]**

[1] Laboratorio de Elucidación y Síntesis en Química Orgánica, ICUAP-BUAP, Puebla 72570, Mexico; eduardo.suarez@correo.buap.mx

[2] Laboratorio de Neuromodulación, Instituto de Fisiología, BUAP, Puebla 72570, Mexico

[3] Laboratorio de Síntesis y Modificación de Productos Naturales, FCQ-BUAP, Puebla 72570, Mexico; jesus.sandoval@correo.buap.mx

[*] Correspondence: jorge.flores@correo.buap.mx (J.F.-H.); alan.carrascoc@correo.buap.mx (A.C.-C.)

**Abstract:** *Ginkgo biloba* (GB) extracts have been used in clinical studies as an alternative therapy for Alzheimer's disease (AD), but the exact bioaction mechanism has not yet been elucidated. In this work, an in silico study on GB metabolites was carried out using SwissTargetPrediction to determine the proteins associated with AD. The resulting proteins, AChE, MAO-A, MAO-B, β-secretase and γ-secretase, were studied by molecular docking, resulting in the finding that kaempferol, quercetin, and luteolin have multitarget potential against AD. These compounds also exhibit antioxidant activity towards reactive oxygen species (ROS), so antioxidant tests were performed on the extracts using the DPPH and ABTS techniques. The ethanol and ethyl acetate GB extracts showed an important inhibition percentage, higher than 80%, at a dose of 0.01 mg/mL. The effect of GB extracts on AD resulted in multitarget action through two pathways: firstly, inhibiting enzymes responsible for degrading neurotransmitters and forming amyloid plaques; secondly, decreasing ROS in the central nervous system (CNS), reducing its deterioration, and promoting the formation of amyloid plaques. The results of this work demonstrate the great potential of GB as a medicinal plant.

**Keywords:** ROS; CNS; AChE; MAO-A; MAO-B; β-secretase; γ-secretase

## 1. Introduction

*Ginkgo biloba* (GB) is a medicinal plant well known to Asian cultures for thousands of years. It is believed that GB originated in the Permian Era, about 250 million years ago [1–3]. It is one of the oldest tree species in the world [1,4] and is the only surviving species of the *Ginkgoaceae* family, which belongs to *Gymnospermae*. It is found in great abundance in China [1,4] and Japan [3]. Its place of origin is believed to be the valleys of Zhejiang, China [1]. Its leaves are green in color, but in autumn they turn golden; the seeds are contained in the Ginkgo "fruit" that is produced by female trees [4]. Its conservation today is due to the preservation of trees in sacred places by Buddhist monks, their resistance to diseases, and their great malleability [1]. In traditional Chinese medicine, it has been used for coughs, skin infections, and gastrointestinal infections due to parasites because it is also very widely cultivated [4].

*G. biloba* extract (EGB761) is widely used worldwide for various diseases, such as memory impairment [1,2,4,5], dementia [1,2], Alzheimer's disease (AD) [1,5,6], peripheral vascular diseases [2,6], glaucoma [2,3], arrhythmias [4], ischemic heart disease [4], cancer [4], diabetes [4], thrombosis [1,4], and cerebrovascular ischemia [6,7], among others. The EGB761 was obtained from dry leaves of *G. biloba* [8]; the identified metabolites are listed in Table 1.

**Table 1.** Reported secondary metabolites of *G. biloba* extract.

| Classification | Secondary Metabolites | References |
|---|---|---|
| Terpenoids | Bilobalide<br>Ginkgolide A<br>Ginkgolide B<br>Ginkgolide C | [1–3,6,7,9,10] |
| Flavonoids | Epicatechin<br>Catechin<br>Apigenin<br>Luteolin<br>Quercetin<br>Kaempferol<br>Isorhamnetin | [1,2,6,10] |
| Carboxylic acids | Protocatechuic acid<br>Caffeic acid<br>Chlorogenic acid<br>p-Hydroxybenzoic acid<br>Vanillic acid<br>p-Coumaric acid<br>Ferulic acid | [6,11] |
| Saccharides | Glucose<br>Rhamnose | [1,6] |

*G. biloba* has been the subject of various reports regarding its effects on the nervous system through the stimulation of blood circulation where the brain is affected, i.e., conditions such as anxiety, stress, lack of concentration, and AD dementia, among others [1]. In an in vitro study, with N2a overexpressed with the human β-amyloid peptide (Aβ) were treated with extracts of *G. biloba*. This study demonstrated an antioxidant effect through a significant increase in total superoxide dismutase (T-SOD) enzymes, catalase (CAT), and glutathione peroxidase (GSH-Px). These data reveal neuroprotection in addition to an effective elimination of induced oxygen free radicals [12]. Particularly, an extract called EGB761 was administered to patients with AD dementia and showed that *G. biloba* extract has an adjuvant effect with acetylcholinesterase inhibitors [13–15].

Another approach to explaining the effect on AD is by using antioxidants since they prevent the formation of free radicals, which lead to the formation of amyloid plaques [1]. EGB761 has been reported to have antioxidant [1,12], anti-inflammatory [1,12], and antidepressant [1] properties, as well as scavenger activity of free radicals [1,12,15] and potent neuroprotective action [1,10,15]. The antidepressant effects can be explained by the reversible inhibition of the two monoamine oxidase isoforms: MAO-A and MAO-B [1]. On the other hand, the anti-inflammatory, antioxidant, and neuroprotective effects are due to the flavonoids and terpenoids of the GB extract [1,14]. Ginkgolide B has shown the ability to inhibit neurotoxicity, induced by β-amyloid peptide (Aβ) [7]. Aβ is generated from the amyloid precursor protein (APP) due to the protease activity of β-secretase and γ-secretase [16].

## 2. Materials and Methods

### 2.1. Studies of Similar Structures

From the metabolites reported in the literature, the targets were obtained using the SwissTargetPrediction platform (STP), SIB Swiss Institute of Bioinformatics, Ginebra, Suiza [17]. In this way, the construction of the frequency diagram was possible, according to the methodology previously reported in [18] for performing a frequency analysis on a set of metabolites from studies. For the analysis of related biological activities, the PassOnline service from Way2Drug was utilized to detect the activities with the highest probability of the set of metabolites [19].

## 2.2. Molecular Docking Studies

Ligands were drawn in 2D Sketcher to minimize Macromodel [20] conformers with OPSL4 and brought to physiological conditions in LigPrep [21] according to previously reported protocol [18]. The proteins were obtained from the Protein Data Bank and coded as 4M0E for AChE [22], 1P0P for BuChE [23], 2Z5Y for MAO-A [24], 2V60 for MAO-B [25], 6C2I for β-secretase [26], and 5FN2 for γ-secretase [27]. Proteins were prepared using Protein Preparation Wizard [28] according to the methodology [18]. Molecular docking was performed on the Glide module [29] with flexibility at the site and using the ligands according to the reported protocol [18].

## 2.3. ADMETx Studies

The prediction of ADME properties was carried out with the minimized structures using the QikProp module [30]. The prediction of types of toxicity and mean lethal dose was carried out by entering the SMILES codes into PassOnline [18] and Gussar [31], respectively.

## 2.4. Extraction and Antioxidant Studies

From batches of 30.0 g of *G. biloba* leaves, differential extraction was performed according to the previously reported protocol [32]. The antioxidant potential of the extracts at a fixed concentration of 100 mg/mL was determined using the techniques of DPPH and ABTS according to previously reported protocols. A total of 1 mL of the stock solution diluted in 1 mL of methanol was used, then 1 mL of the DPPH free-radical solution was added, homogenized, incubated, and isolated from light for 1 h. Gallic acid (0.4836 mg/mL) was used as a control [33].

## 3. Results and Discussion

The in silico results were analyzed according to the interest related to neurodegenerative diseases. The first results obtained from STP were the proteins that interact with the *G. biloba* metabolites shown in Table 1. Later, PassOnline showed results of the biological activity of these metabolites; a study of the different ADME properties of each structure in Table 1 was also carried out. Based on these results, the reference proteins, structures, and drugs were delimited for the next step, the molecular docking. Finally, the DPPH and ABTS tests were launched to determine the potential antioxidant activity of the *G. biloba* differential extracts.

## 3.1. Analyses for Similar Structures

From STP it was found that 218 possible targets have at least one interaction with a secondary metabolite of interest. The percentage frequency graph of Figure 1 shows those proteins with high frequency. Figure 1 shows that the maximum number of interactions that occurred was 13, corresponding to carbonic anhydrase 12 (CA12) with a frequency of interest of 65%. However, the isoforms of CA IX, I, IV, XIV, VI, VII, and III had an interesting frequency, greater than 50%. In the case of the CA-VA (CA5A), it had a frequency of 50%, and finally CA XIII had a 45% frequency of interest. Carbonic anhydrases (CAs) belong to the family of metalloenzymes that contain a zinc molecule in their active site [34]. CAs are enzymes that catalyze several non-hydrolytic, non-oxidative elimination reactions or the lysis of a substrate [34], with great importance in the normal physiological processes of the body.

Figure 1 shows four proteins (APP, AChE, MAO-A, and MAO-B) of interest associated with neurodegenerative diseases (NDs), all of them related to AD (Table 2). Considering this relationship with AD, we decided to work with these proteins and with butyryl-cholinesterase (BuChE), a cholinesterase useful as an indicator of poisoning by organophosphates or pesticides [35]. Some important roles have been ascribed to the amyloid precursor protein (APP) in synaptic processes and transcellular synaptic adhesion [36]. APP, as its name suggests, is the precursor of β-amyloid peptide (Aβ) [37], derived from proteolysis of β- and γ-secretases [36–38], giving rise to the characteristic senile plaques of AD [39].

Being able to inhibit these two secretases, APP has revealed a challenge in pharmacology due to the poor pharmacokinetic properties of the drugs and their selectivity [38]. The objective of being able to inhibit both β- and γ-secretases is to prevent the two active protease complexes from hydrolyzing the APP transmembrane domains for the formation of Aβ and with it the formation of fibrils and accumulation of senile plaques [36–38,40]. Table 2 summarizes the NEs related to each of the proteins and their known inhibitors.

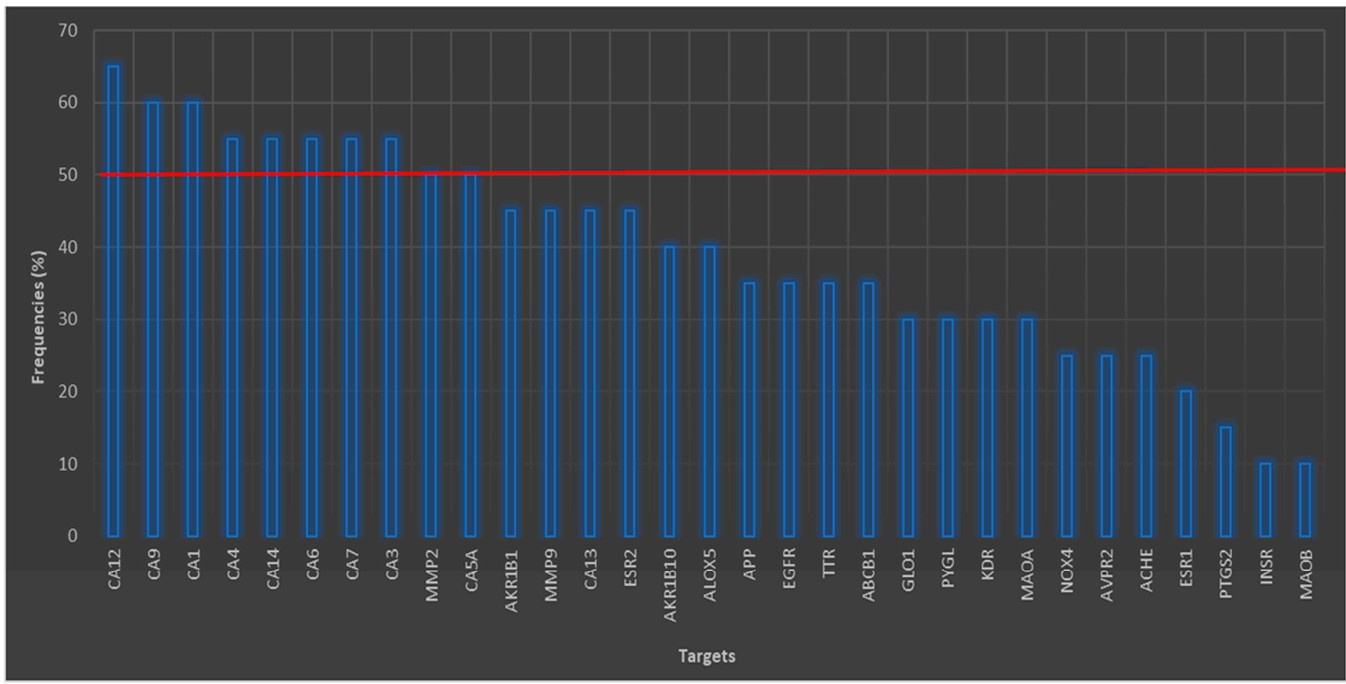

**Figure 1.** Targets obtained through structural analyses of similar GB secondary metabolites.

**Table 2.** Proteins of interest related to neurodegenerative diseases.

| Proteins | Diseases | Inhibitors | References |
|---|---|---|---|
| MAO-A | Alzheimer's disease<br>Glioma<br>Depressive disorders | Moclobemide<br>Clorgyline<br>Tranylcypromine | [10,16,41–44] |
| MAO-B | Alzheimer's disease<br>Parkinson's disease<br>Depressive disorders | Rasagiline<br>Selegiline<br>Deprenil | [10,16,42–45] |
| AChE | Alzheimer's disease<br>Parkinson's disease<br>Huntington's disease<br>Depressive disorders<br>Myasthenia gravis | Galantamine<br>Huperzine A<br>Rivastigmine<br>Neostigmine<br>Donepezil | [46–50] |
| BuChE | No physiological functions<br>Alzheimer's disease<br>Myasthenia gravis | Neostigmine<br>Huperzine A<br>Rivastigmine | [35,49,51] |
| β-Secretase | Alzheimer's disease | GRL-8234<br>CTS-21166<br>Verubecestat | [40,52,53] |
| γ-Secretase | Alzheimer's disease | Avagacestat<br>Semagacestat | [40,52] |

The results of PassOnline indicated that some of the GB metabolites such as bilobalide, ginkgolide A, ginkgolide B, and ginkgolide C have probability of antioxidant activity (0.960 or greater), this activity being of interest. Likewise, these four metabolites have potential activity for inhibition of platelet aggregation and glycine receptor antagonists. Ginkgolides A, B, and C showed activity for treatment of neurodegenerative diseases, as well as activity for the treatment of AD. A point to consider is that only ginkgolides B and C had a probability of activity greater than 0.794 as Aβ antagonists. In the case of isorhamnetin, kaempferol,, and quercetin, they had a probable activity as inhibitors of the expression hypoxia-inducible factor 1-alpha (HIF1-$\alpha$), which is the main mediator of various signaling pathways in breast cancer when there is tumor hypoxia [54]. The other potential activities are not related to neurodegenerative diseases.

### 3.2. Molecular Docking Studies

The molecular coupling studies are shown in Table 3, where it can be observed that some metabolites such as ginkgolides A, B, and C did not interact with MAO-A, MAO-B, AChE, and $\gamma$-secretase proteins. As with AChE, 12 of the metabolites did not interact with the protein, given their molecular size and the high specificity of the active site. On the other hand, both BuChE and $\beta$-secretase were the only proteins that did interact with all the metabolites. Table 3 also shows that the molecular docking studies were performed with endogenous ligands and with known commercial inhibitors to compare the inhibition potential.

**Table 3.** Molecular docking studies of proteins related to the CNS.

| Secondary Metabolite | MAO-A | MAO-B | AChE | BuChE | β-Secretase | γ-Secretase |
|---|---|---|---|---|---|---|
| Bilobalide | NI | NI | NI | −6.056 | −4.694 | −0.110 |
| Ginkgolide A | NI | NI | NI | −6.559 | −5.413 | NI |
| Ginkgolide B | NI | NI | NI | −6.350 | −6.132 | NI |
| Ginkgolide C | NI | NI | NI | −6.068 | −6.230 | NI |
| Isorhamnetin | −6.223 | −6.588 | NI | −6.328 | −5.835 | −5.763 |
| Kaempferol | −9.127 | −6.636 | NI | −6.160 | −6.445 | −5.955 |
| Quercetin | −7.872 | −6.626 | NI | −7.451 | −6.244 | −4.668 |
| Apigenin | −8.698 | −6.663 | NI | −6.047 | −6.359 | −5.938 |
| Vanillic acid | −6.314 | −6.227 | −6.038 | −5.990 | −4.505 | −4.754 |
| Caffeic acid | −5.089 | −5.891 | −4.843 | −5.715 | −4.564 | −5.445 |
| Catechin | −7.287 | NI | NI | −6.321 | −7.345 | −4.652 |
| Epicatechin | NI | NI | NI | −7.984 | −5.969 | −4.753 |
| Chlorogenic acid | −4.548 | −6.302 | NI | −5.773 | −4.571 | −4.721 |
| Ferulic acid | −5.650 | −6.346 | −5.373 | −5.669 | −4.805 | −4.717 |
| Luteolin | −6.587 | −6.667 | NI | −7.114 | −5.539 | −5.344 |
| p-Coumaric acid | −4.642 | −5.500 | −4.354 | −5.751 | −4.753 | −4.647 |
| p-Hydroxybenzoic acid | −6.510 | −5.902 | −6.209 | −6.294 | −4.868 | −4.828 |
| Protocatechuic acid | −6.767 | −6.079 | −5.815 | −6.488 | −4.854 | −4.718 |
| (L)-Rhamnose | −5.859 | −5.901 | −5.780 | −5.822 | −4.892 | NI |
| (D)-Glucose | −5.498 | −5.627 | −5.695 | −6.941 | −4.848 | −5.801 |
| Harmine | −6.916 | - | - | - | - | - |
| Phenelzine | - | −5.766 | - | - | - | - |
| M30 | −5.956 | −6.104 | - | - | - | - |
| Acetylcholine | - | - | −3.546 | - | - | - |
| BuCh | - | - | - | −4.341 | - | - |
| Donezepil | - | - | −4.509 | - | - | - |
| Rasagiline | - | −6.060 | −5.145 | - | - | - |
| Physostigmine | - | - | - | −5.874 | - | - |
| Verubecestat | - | - | - | - | −4.298 | - |
| Semagacestat | - | - | - | - | - | −4.446 |

NI = no interaction.

3.2.1. Monoamine Oxidase Pathway

In the case of the monoamine oxidase (MAO) pathway, the drugs phenelzine and M30 were taken as a reference because they are responsible for inhibiting the two isoforms A and B. For MAO-A, the drug M30 had a binding coupling energy of −5.956 kcal/mol, while for MAO-B it was −6.104 kcal/mol. In the case of the drug phenelzine, it only had an interaction with MAO-B. epicatechin had no interaction in both isoforms, while catechin had no interaction with MAO-B. In this MAO pathway, isoform A had better coupling energy towards GB metabolites than isoform B in seven cases, while isoform B had better coupling energy in eight cases. Therefore, in the MAO pathway, the isoform that tends to be inhibited with greater intensity is MAO-B due to the number of better coupling energies compared to the A isoform. The next study was the comparison of the binding coupling energies of the GB metabolites and the reference drug M30 with the two isoforms. In the case of MAO-A, there were 10 metabolites that had a higher coupling energy than the drug; for case of MAO-B, only eight metabolites had a better coupling energy than the same drug. In the case of isoform A, only three metabolites had a better coupling energy than the drug harmine. In the case of isoform B, 11 metabolites had better couple energies than the drug phenelzine. In this case there were nine metabolites with better coupling energies; a curious result with Drug M30 is that it also had an interaction with AChE and had a better energy than acetylcholine, caffeic acid and *p*-coumaric acid.

The amino acid residues reported for the inhibition of MAO-A are Cys 323, Ile 180, Ile 335, Leu 97, Leu 337, Glu 216, Met 350, Phe 208, Phe 352, Ser 209, Thr 336, Tyr 69, Tyr 197, Tyr 407, and Tyr 444 [44,55]. In contrast, the amino acid residues reported in this article, from the crystal data, show that Asn 181, Cys 323, Gln 215, Ile 325, Ile 335, Leu 337, Phe 208, Phe 352, Tyr 69, Tyr 407, and Tyr 444 interact with the drug harmine [24]. In Figure 2, it is observed that M30, harmine, and quercetin interact with Cys 323, Ile 335, Leu 337, Phe 208, Phe 352, Tyr 69, Tyr 407, and Tyr 444 residues that were previously described to be residues where enzyme inhibition takes place. The difference between the three mentioned molecules is the binding coupling energy (BCE), which corresponds to −5.956, −6.916 and, −7.872 kcal/mol, respectively, with quercetin being the structure that has the greatest coupling to the protein and is capable of interacting with amino acids reported for MAO-A inhibition. Finally, kaempferol had a better BCE than all previous structures and only interacted with five of the residues reported for inhibition: Cys 323, Ile 335, Leu 337, Phe 208, and Phe 352, without interaction with tyrosine residues. This may be because kaempferol is able to build a π–π stacking interaction with Phe 208 and two hydrogen bonds with residues Ala 111 and Asn 181. However, this result is interesting because quercetin had the same or more interactions than kaempferol, but with a lower EA.

The amino acid residues for MAO-B inhibition are Cys 172, Gln 65, Gln 206, Ile 198, Ile 199, Ile 316, Leu 167, Leu 171, Phe 103, Phe 168, Pro 104, Trp 119, Tyr 326, and Tyr 435 [44,56,57]. On the other hand, safinamide, which was the molecule used as the inhibitor of MAO-B, had the following reported interaction with amino acid residues: Cys 172, Gln 206, Ile 199, Ile 316, Leu 171, Phe 103, Tyr 60, Tyr 326, and Tyr 398 [25].

Figure 3 shows that the molecule with the most interactions with the amino acid residues (Cys 172, Gln 206, Ile 199, Ile 316, Leu 171, Phe 103, Phe 168, Tyr 60, Tyr 326, Tyr 398, and Tyr 435), reported for the inhibition of MAO-B, was luteolin. Also, it was the molecule that had the best BCE of −6.636 kcal/mol. This could be due to the number of interactions and because the Tyr 326 residue forms a π–π stacking interaction and a hydrogen bond in different parts of luteolin. M30, which was used for the two isoforms, had the second-highest number of interactions with the amino acids reported for enzyme inhibition: Gln 206, Ile 199, Ile 316, Leu 171, Phe 103, Phe 168, Tyr 60, Tyr 326, Tyr 398, and Tyr 435. This last residue forms a hydrogen bond with M30. The kaempferol metabolite interacted with nine of the amino acids for inhibition: Cys 172, Gln 206, Ile 199, Ile 316, Leu 171, Phe 103, Phe 168, Tyr 60, and Tyr 326. Tyr 326 residue forms an interaction of the π–π stacking type; this molecule has the second-best BCE of −6.636 kcal/mol. An interesting similarity between kaempferol and luteolin is that both molecules form the π–π

stacking interaction with Tyr 326, so it can be assumed that this interaction is the one that allows a higher binding force with respect to the other molecules that do not present this interaction. Finally, phenelzine had the lowest BCE of −5.766 kcal/mol and the lowest number of interactions with the inhibiting amino acids: Cys 172, Gln 206, Ile 199, Ile 316, Leu 171, Phe 103, Phe 168, and Tyr 326. Both Ile 199 and Tyr 326 form a hydrogen bond, which may respond to the lower affinity they have with respect to the other structures.

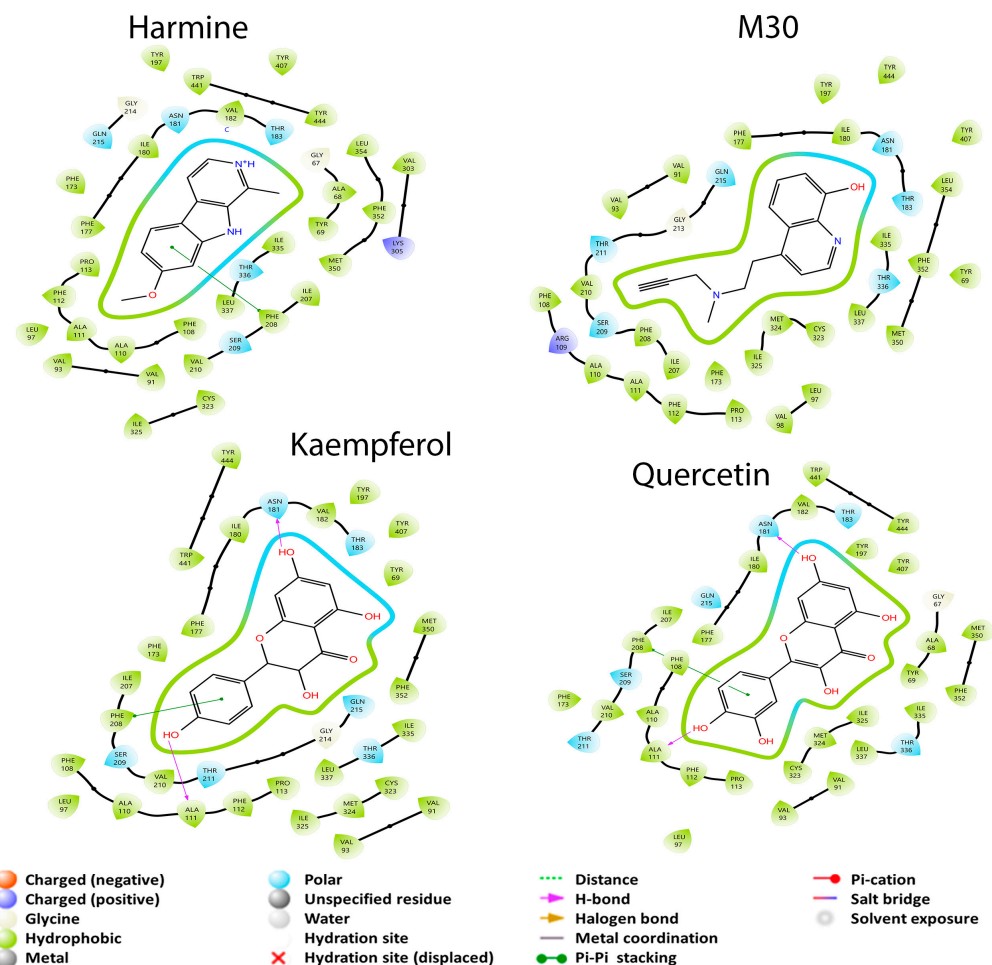

**Figure 2.** 2D diagrams of the interactions of the dual drug M30, the MAO-A specific drug harmine, and the best GB metabolites (quercetin and kaempferol) with MAO-A.

### 3.2.2. Enzymatic Cholinergic Pathway

Eight metabolites presented interaction with AChE, while with BuChE all metabolites had interaction. Acetylcholine and butyrylcholine were used as reference endogenous substrates, as shown in Figure 4. All the coupling energies of the GB metabolites that interacted with the two enzymes were greater than those of their reference their reference ligands. For this study, this is not the most convenient, since one of the main pathways or mechanisms of action involved in AD is the participation of AChE in the CNS. For BuChE, its function in the body is not yet well defined; therefore, its large number of interactions with GB metabolites and their effects are currently unknown. From these data, it can be inferred that GBS does not exert its effect in AD treatment through the cholinergic pathway.

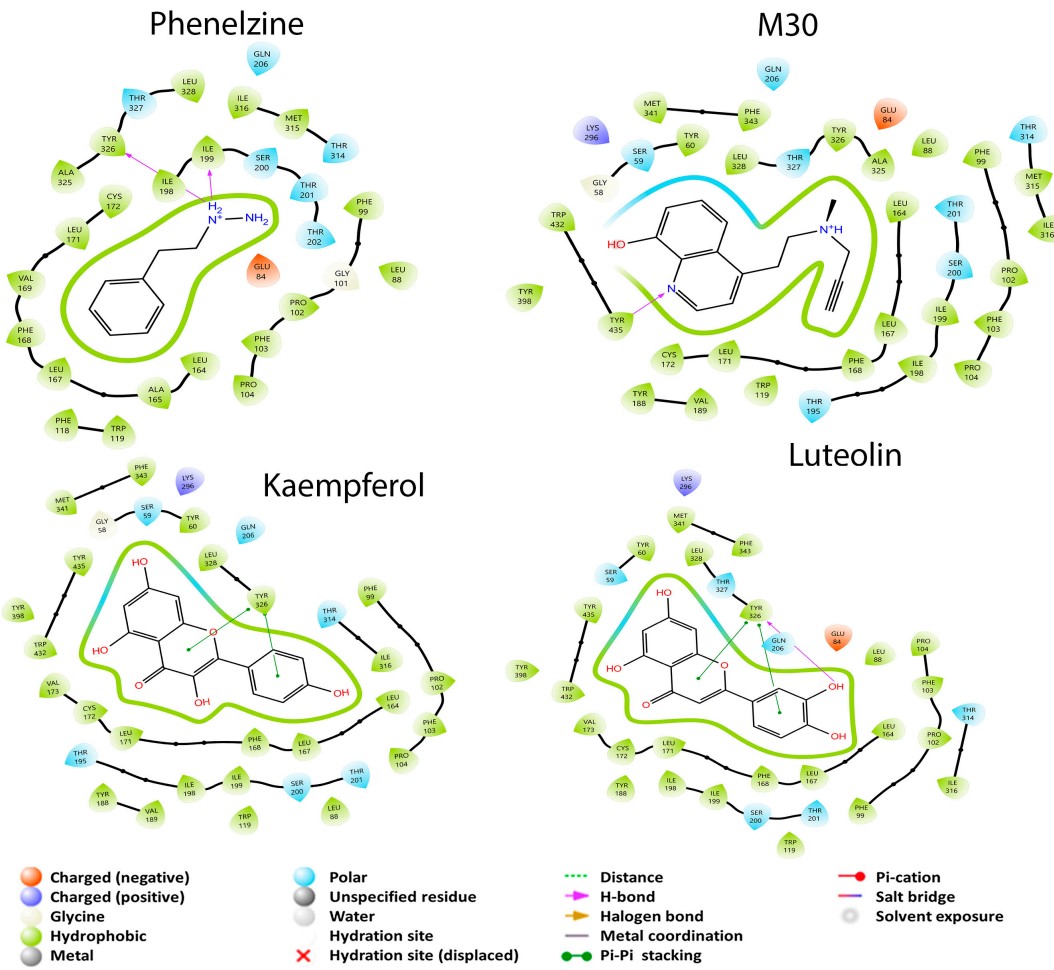

**Figure 3.** 2D diagrams of the interactions of the dual drug M30, the MAO-B specific drug phenelzine, and the best GB metabolites (luteolin and kaempferol) with MAO-B.

The endogenous ligand of this enzyme is butyrylcholine, which had a BCE of −4.341 kcal/mol; according to Nicolet et al. [23], Ser 198 is the residue responsible for the catalytic activity of the enzyme. Some other important amino acids for the enzyme activity are Ala 199, Glu 197, Gly 116, Gly 117, His 438, Met 437, Phe 329, Ser 287, and Trp 231. On the other hand, Shakil [58] has found that Ala 328, Gly 116, His 438, Met 437, Phe 329, Trp 82, Trp 430, Tyr 332, and Tyr 440 amino acids are responsible for the tested inhibitor interactions, having two residues in common. Figure 4 shows that butyrylcholine interacted with thirteen of the previously mentioned amino acids in common with the other study structures, including Ser 198: Ala 199, Ala 328, Glu 197, Gly 117, Hip 438, Met 437, Phe 329, Ser 198, Ser 287, Trp 82, Trp 430, Tyr 440, and Val 288; however, we can see that the Hip 438 residue forms a hydrogen bridge while Glu 197 forms a salt-bridge-type interaction. Physostigmine is an inhibitor used for this enzyme and has a BCE of −5.874 kcal/mol, like the endogenous ligand. Physostigmine is capable of binding to the same thirteen residues as the endogenous ligand, including the hydrogen bond interaction of the Hip 438 residue. The difference with butyrylcholine is that a salt bridge interaction does not occur here.

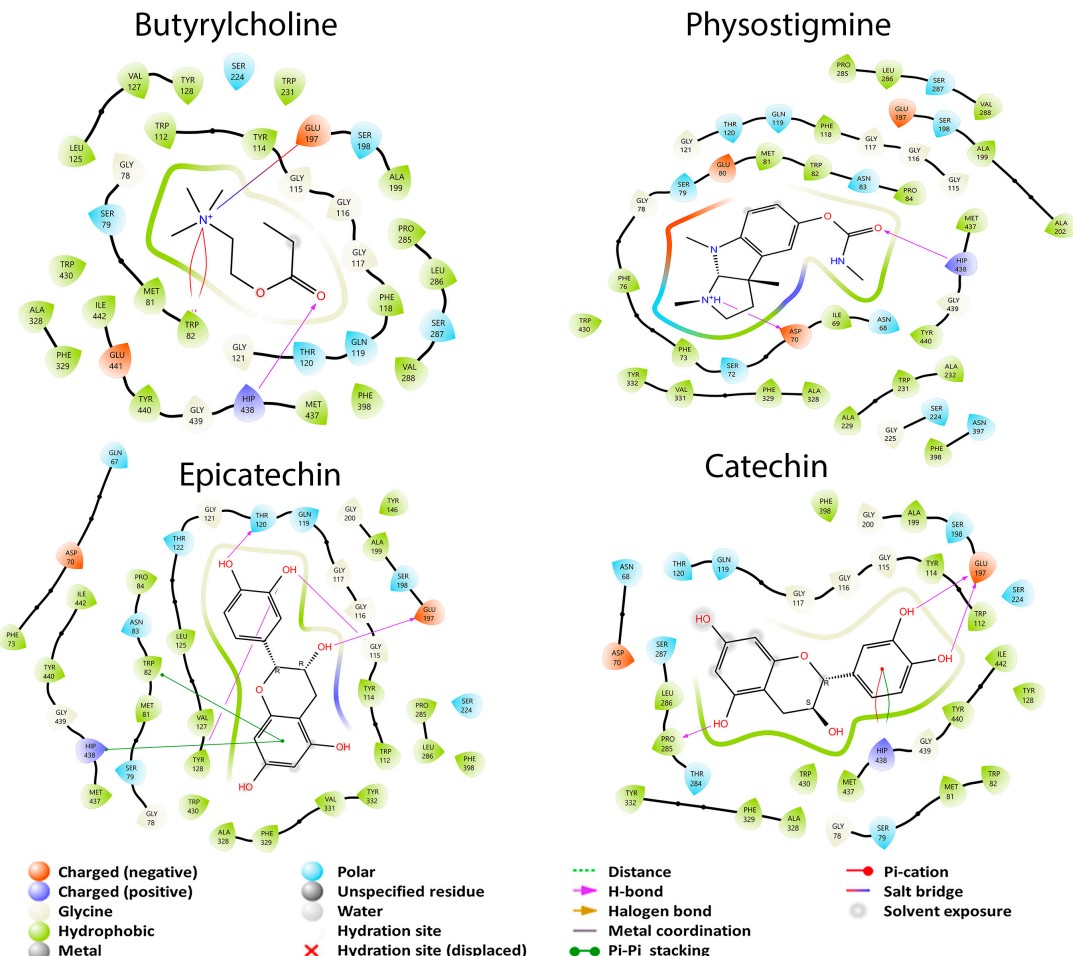

**Figure 4.** 2D diagrams of the interactions of the endogenous ligand butyrylcholine, the drug physostigmine and the best GB metabolites (catechin and epicatechin) with BuChE.

Catechin had a better BCE (−6.321 kcal/mol) than butyrylcholine and physostigmine, showing twelve interactions in common with the previous structures: Ala 199, Ala 328, Glu 197, Gly 117, Hip 438, Met 437, Phe 329, Ser 198, Ser 287, Trp 82, Trp 430, and Tyr 440. Unlike the previous structures, catechin did not interact with the Val 288 residue and has a double interaction with the residue Hip 438 with a π–π stacking and a π–cation interaction. Also, it presented a hydrogen bond with the Glu 197 residue. These last interactions could be responsible for the catechin having a better BCE than butyrylcholine and physostigmine. Epicatechin, which is the enantiomer of catechin, had a better BCE of −7.984 kcal/mol than all previous molecules. In this case, this metabolite presented 11 interactions in common with the other structures: Ala 199, Ala 328, Glu 197, Gly 117, Hip 438, Met 437, Phe 329, Ser 198, Trp 82, Trp 430, and Tyr 440. Like catechin, this molecule had a hydrogen-bond-type interaction with the Glu 197 residue and a π–π stacking interaction with Hip 438. We can observe that the isomerism of these last two molecules does affect AD with BuChE and that the Hip 438 residue that forms a π–cation interaction may result in catechin having a BCE inferior to that of epicatechin. We can observe that at least eleven amino acid residues share these structures in common and mainly with the Ser 198 residue, which is important for the BuChE catalytic site.

AChE has several subsites such as the oxyanion region with residues: Ala 204, Gly 121, and Gly 122; the anionic subsite: Ile 451, Glu 202, Gly 448, Trp 86, and Tyr 133; and the peripheral anion site (PAS) that comprises Asp 74, Ser 125, Trp 286, Tyr 72, Tyr 124, Tyr 337, and Tyr 341 and provides a binding site for allosteric modulators, inhibitors, and other residues of the omega group: Asn 87, Pro 88, and Thr 83 [59–61]. The difference of BCE

between ACh and AChE was −3.546 kcal/mol, and according to the catalytic triad it is represented by Ser 203, His 447, and Glu 334 residues found in the active site of the protein. Figure 5 shows that ACh presents an interaction with Ala 204, Asn 87, Asp 74, Glh 202, Gly 121, Gly 122, Pro 88, Ser 125, Thr 83 Trp 86, Trp 286, Tyr 72, Tyr 124, Tyr 133, Tyr 337, and Tyr 341 and with two residues of the catalytic triad (Ser 203 and Hip 447). These last two residues are responsible for the biological activity of the enzyme, and that is why ACh was taken as a reference. On the contrary, donezepil is a specific and reversible inhibitor of acetylcholinesterase. It has a BCE of −4.509. Remarkably, this result shows that donezepil does not bind to any residue of the AChE catalytic triad; it only presents a hydrophobic interaction of those reported in the literature and in common with ACh: Trp 236. Rasagiline is an irreversible inhibitor of MAO used for early PD and interestingly had an affinity for AChE with a BCE of −5.145 kcal/mol. It interacted with two residues of the catalytic triad (Hip 447 and Ser 203) but also had interactions in common with ACh and GB metabolites; with donezepil it did not have any in common: Ala 204, Asn 87, Asp 74, Glh 202, Gly 121, Gly 122, Gly 148, Pro 88, Ser 125, Thr 83, Trp 86, Tyr 72, Tyr 124, Tyr 133, Tyr 337, and Tyr 341, and presented a π–π-stacking-type interaction at residue Trp 86.

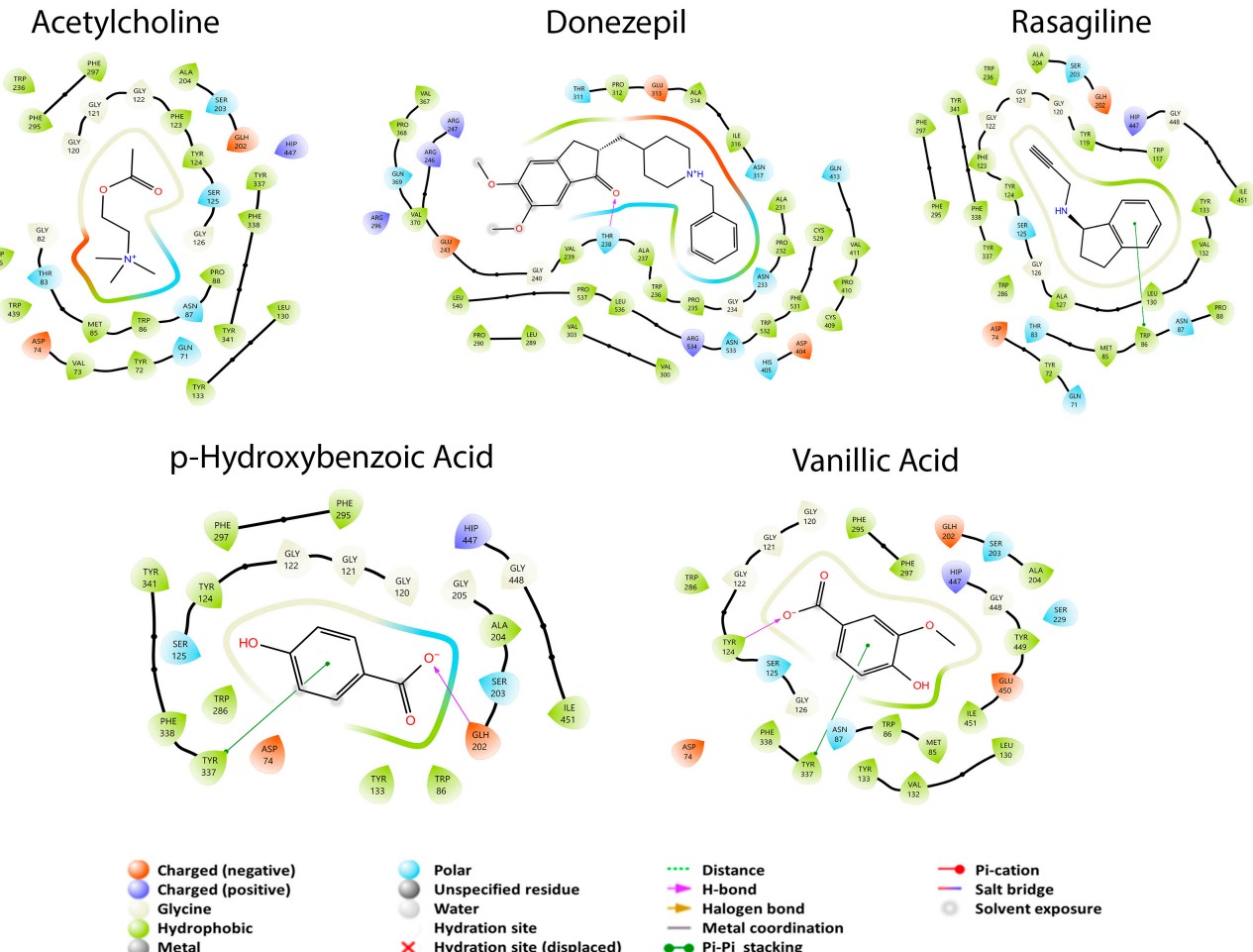

**Figure 5.** 2D diagrams of the interactions of the endogenous ligand acetylcholine, the drug Rasagiline, the inhibitor donezepil, and the best GB metabolites (p-hydroxybenzoic acid and vanillic acid) with AChE.

Finally, p-hydroxybenzoic acid had a BCE of −6.209 kcal/mol, being the molecule that had the best BCE associated with AChE. This metabolite also interacted with residues Hip 447 and Ser 203 of the catalytic triad of the enzyme but also showed interactions with other residues such as Ala 204, Asp 74, Glh 202, Gly 121, Gly 122, Gly 448, Ile 451, Ser 125, Trp

86, Trp 286, Tyr 124, Tyr 133, Tyr 337, and Tyr 341. It also presented a hydrogen-bond-type interaction with Glh 202 and a π–π stacking interaction with Tyr 337. From the latter results it can be pointed out that the Tyr 337 residue that has a π–π stacking interaction is one of those responsible for the fact that vanillic acid and p-hydroxybenzoic acid have a better BCE than the rest of the molecules. The interaction also influences the hydrogen bonding that each of these metabolites presents.

### 3.2.3. APP Secretase Regulation

Finally, there are two closely related proteins needed to produce APP that will give rise to Aβ, β-secretase and the γ-secretase. In this case, the γ-secretase did not interact with rhamnose and the three ginkgolides A, B, and C, while β-secretase did interact with all GB metabolites. In the case of γ-secretase, it only had four better coupling energies with the metabolites described with respect to β-secretase, while the latter had sixteen better coupling energies with the metabolites. Verubecestat is a drug for β-secretase which had a coupling energy of −4.298 kcal/mol. In the case of this protein, the twenty metabolites had a better coupling energy than the drug. In the case of γ-secretase, the drug used was Semagacestat with a coupling energy of −4.446 kcal/mol; here, only 15 GB metabolites had a better coupling energy than that observed with the reference drug. From this pathway, it can be detected that the protein that tends to be more likely to be inhibited is β-secretase due to the number of interactions it has with the metabolites and their coupling energies. The molecule used as the ligand in the crystallized protein was cyclopropyl 2 (cyclopropyl-3-amino-2,4-oxazine) [26], showing important residues in the molecular coupling: Asp 32, Asp 228, Gly 230 (possibly the active site), Arg 128, Thr 232, Trp 76, Trp 115, Tyr 71, and Tyr 198. It is proposed that the residues of Phe 108, Ser 35, and Thr 231 are of importance since they are probably the residues of an allosteric site of the protein [26]. Figure 6 shows that Verubescestat is a molecule under current investigation as a β-secretase inhibitor. The BCE between the protein and this structure was −4.298 kcal/mol due to an interaction with three amino acid residues in common with the best metabolites of GB: luteolin, kaempferol, quercetin, and catequin (Phe 108, Ser 35, and Thr 231), residues that in the article by Low et al. are considered important for the allosteric site [26].

Verubescestat has interactions with Arg 127, Arg 234, Asp 31, Asp 227, Gln 11, Gly 12, Gly 33, Gly 229, Ile 109, Ile 117, Ile 125, Ile 225, Leu 29, Phe 107, Ser 34, Thr 230, Thr 328, Trp 75, Trp 114, Tyr 70, Tyr 197, and Val 331 [62]. Luteolin had a BCE of −5.539 kcal/mol and presented two hydrogen bond interactions (Gly 229 and Tyr 197). Kaempferol had a BCE of −6.445 kcal/mol and presented three hydrogen bond interactions (Asp 227, Gly 33, and Phe 107). The Phe 107 residue may be responsible for a better BCE of kaempferol than the other ones. Quercetin had a BCE of −6.244 kcal/mol and presented the same hydrogen bond interactions as kaempferol; however, quercetin interacted with other residues, producing a better BCE (Gly 10 and Val 68). Lastly, catechin had the best BCE (−7.345 kcal/mol) of all molecules that interacted with β-secretase, and it possesses the same three hydrogen bonding interactions as quercetin and kaempferol. However, compared to these last two metabolites, catechin did not present interaction with Gly 10, Ser 34, and Ser 112.

For γ-secretase, the dipeptide inhibitor (DAPT) was used as the ligand, which interacts with residues: Asp 257, Asp 385, Glu 280, Gly 384, His 163, Ile 143, Leu 166, Met 146, Met 233, Phe 283, Ser 169, and Trp 165 [27]. Semagacestat is under investigation as a treatment for AD through γ-secretase inhibition [40]. This last experimental molecule had a BCE of −4.446 kcal/mol with the study protein. Figure 7 shows that the four structures: semagacestat, luteolin, kaempferol, and quercetin have in common that none of them had any interaction with any of the previously mentioned residues.

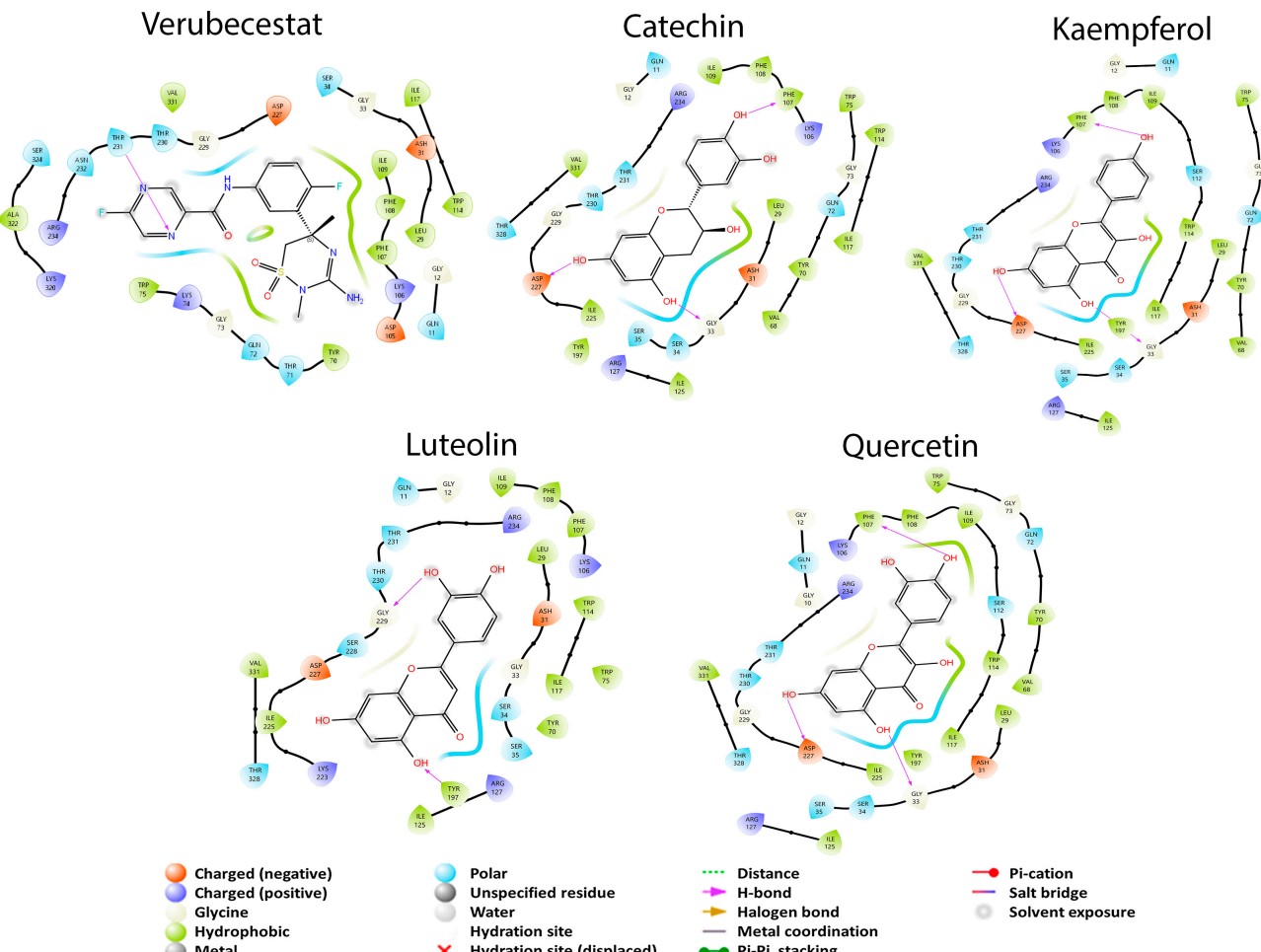

**Figure 6.** 2D diagrams of the interactions of verubecestat, catechin, luteolin, kaempferol and quercetin (the four best GB metabolites) with β-secretase.

Although none of the four molecules in Figure 7 interacted with the reported residues, they did have common interactions with Ala 141, Asp 140, Asp 152, Gly 143, Gly 145, Gly 148, Gly 151, Hid 150, Ile 149, Phe 157, Pro 144, and Ser 153. Semagacestat had a BCE of −4.446 kcal/mol and had in common with luteolin and kaempferol the formation of hydrogen bonds with Asp 152 and Gly 148. However, semagacestat differs from the two metabolites due to the formation of a hydrogen bond with Ile 149. Luteolin had a better BCE (−5.344 kcal/mol) than the experimental molecule and formed hydrogen bonds with Asp 140 and Hid 150. In the case of kaempferol, the metabolite had the best BCE (−5.955 kcal/mol) towards the study protein and had hydrogen bond interactions with Asp 140 and Ile 149. Finally, quercetin had a BCE of −4.668 kcal/ mol and had hydrogen bonding interactions with Gly 145 and Hid 150. This can give an idea about the difference in BCE values; it is possible that the hydrogen bond interaction with Asp 140 is responsible for a better BCE in the metabolites luteolin and kaempferol than in the others. On the other hand, it possible that none of the molecules is bound to the active site of the protein, due to the absence the absence of a clear and concise report about which residues are responsible for the activity; however, the similarity of interactions with amino acids can give the idea that they are bound to an allosteric site.

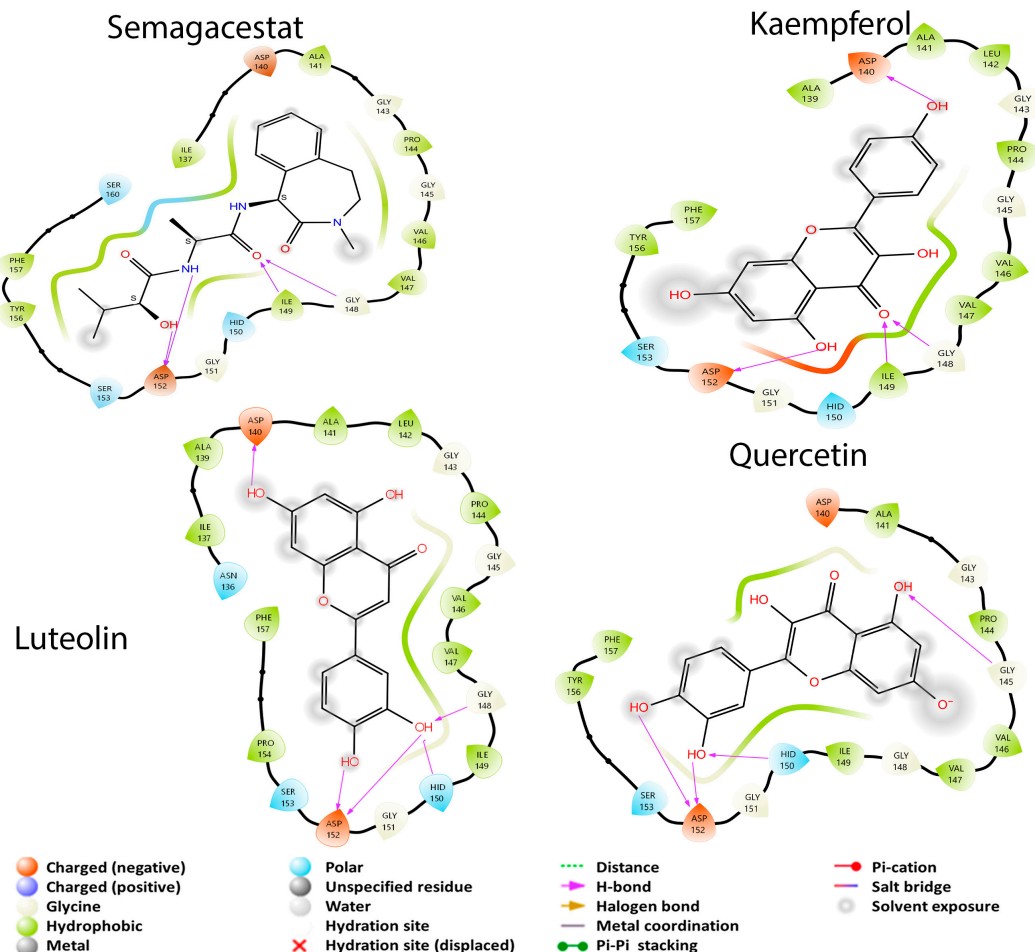

**Figure 7.** 2D diagrams of the interactions of semagacestat, luteolin, kaempferol, and quercetin (the best GB metabolites) with γ-secretase.

### 3.3. ADMETx Studies

The pharmacokinetic studies of the substances that enter the body is an important part a drug's development, so changes occurring inside an organism must be discussed. Studies of absorption, distribution, metabolism, and excretion (ADME) the most commonly used to understand these processes in the body [17].

Table 4 shows that none of the secondary metabolites exceed a MW greater than 500 g/mol and that they fall within an adequate range of lipophilicity according to [63]. An important pharmacokinetic parameter to be determined is whether the metabolites can cross the BBB; however, there are transport methods for flavonoids into the CNS that stand out as passive transport and through channels [64–66]. Particularly, luteolin, p-hydroxybenzoic acid, and protocatechuic acid can cross this barrier, not depending on alternative mechanisms. According to their solubility, it is important to manage molecules having a Log S less than 6 mg/mL, being true for all compounds. It is important to determine the GI absorption values for each metabolite if oral administration is required. The bioavailability of the molecules in the body should be known as well as the best route of administration; for this purpose, the Rule of Five (Lipinski) and the Rule of Three (Veber) were determined. The latter represent adequate properties for a compound to have good oral viability; in the case of GB metabolites, ginkgolide B and ginkgolide C, ferulic acid, rhamnose, and glucose had low GI absorption, while the rest had high absorption. Of all the metabolites, Table 4 shows that only ferulic acid complies with the Veber and Lipinski rules. In the case of isorhamnetin, kaempferol, vanillic acid, and chlorogenic acid, they

complied with the Rule of Three, while ginkgolide C complied with the Rule of Three, and the rest of the molecules did not comply with any rule.

**Table 4.** ADME parameters * for GB metabolites.

| Secondary Metabolite | MW (g/mol) | Lipophilicity Log P | BBB | Water Solubility Log S | GI Absorption | Rule of Three | Rule of Five |
|---|---|---|---|---|---|---|---|
| Bilobalide | 326.3 | 0.07 | No | −1.12 | High | 0 | 0 |
| Ginkgolide A | 408.4 | 0.61 | No | −1.59 | High | 0 | 0 |
| Ginkgolide B | 424.4 | −0.2 | No | −0.77 | Low | 0 | 0 |
| Ginkgolide C | 440.4 | −0.98 | No | 0.06 | Low | 0 | 1 |
| Isorhamnetin | 290.27 | 0.85 | No | −2.14 | High | 1 | 0 |
| Kaempferol | 290.27 | 0.85 | No | −2.14 | High | 1 | 0 |
| Quercetin | 270.24 | 2.11 | No | −4.4 | High | 0 | 0 |
| Apigenin | 286.24 | 1.73 | No | −3.82 | High | 0 | 0 |
| Vanillic acid | 302.24 | 1.23 | No | −3.24 | High | 1 | 0 |
| Caffeic acid | 286.24 | 1.58 | No | −3.82 | High | 0 | 0 |
| Catechin | 316.26 | 1.65 | No | −3.94 | High | 0 | 0 |
| Epicatechin | 154.12 | 0.65 | No | −0.6 | High | 0 | 0 |
| Chlorogenic acid | 180.16 | 0.93 | No | −0.71 | High | 1 | 0 |
| Ferulic acid | 354.31 | −0.38 | No | 0.4 | Low | 1 | 1 |
| Luteolin | 138.12 | 1.05 | Yes | −1.17 | High | 0 | 0 |
| p-Coumaric acid | 168.15 | 1.08 | No | −1.32 | High | 0 | 0 |
| p-Hydroxybenzoic acid | 164.16 | 1.26 | Yes | −1.28 | High | 0 | 0 |
| Protocatechuic acid | 194.18 | 1.36 | Yes | −1.42 | High | 0 | 0 |
| L-Rhamnose | 180.16 | −2.26 | No | 2.45 | Low | 0 | 0 |
| D-Glucose | 164.16 | −1.5 | No | 1.9 | Low | 0 | 0 |

* Different pharmacokinetic properties of GB metabolites are shown. BBB: blood–brain barrier permeant, GI absorption: gastrointestinal absorption, MW: molecular weight.

### 3.4. Extraction and Antioxidant Studies

The formation of free radicals due to the activity of nNOS is directly related to the fragmentation of amyloids generating the aggregation of plaques, which is why the use of antioxidants is sought, given that alterations provoked by some proteins in the nervous system are responsible for the pathophysiology of AD; however, other causes of nerve cell deterioration are possible, such as the presence of oxidative stress. Excess of reactive oxygen species (ROS) and bioactive metals promote the production of Aβ and hyperphosphorylated tau [64]. Even the previously studied extract has demonstrated antioxidant activity because EGB761 has shown strong oxygen free-radical scavenging as well as anti-inflammatory and neuroprotective activities [1,12,67–69].

Therefore, it was of interest to study the antioxidant potential of the various extracts obtained. In Figure 8, the results of the inhibition percentages of the DPPH and ABTS tests of each GB extract (eight different solvents) can be observed, ranging from a low to high polarity. The methanol extract had the highest percentage of inhibition (84.7%), while extracts with acetone (72.4%), ethanol (67.7%), hexane (67.4%), ethyl ether (67.2%), and ethyl acetate (55%) can also be seen. The other extracts had an inhibition percentage of less than 50%: water (45.3%) and dichloromethane (41.6%) in the case of the DPPH test. In the ABTS test, it is observed that the extract with ethanol had the highest percentage of inhibition (86.7%), followed by the extract with ethyl acetate (81.9%), and ethyl ether (60.9%). The rest of the extracts had an inhibition percentage less than 50%: DCM (48.5%), acetone (46.2%), methanol (43.6%), water (27.3%), and hexane that showed the lowest percentage of inhibition (18.5%).

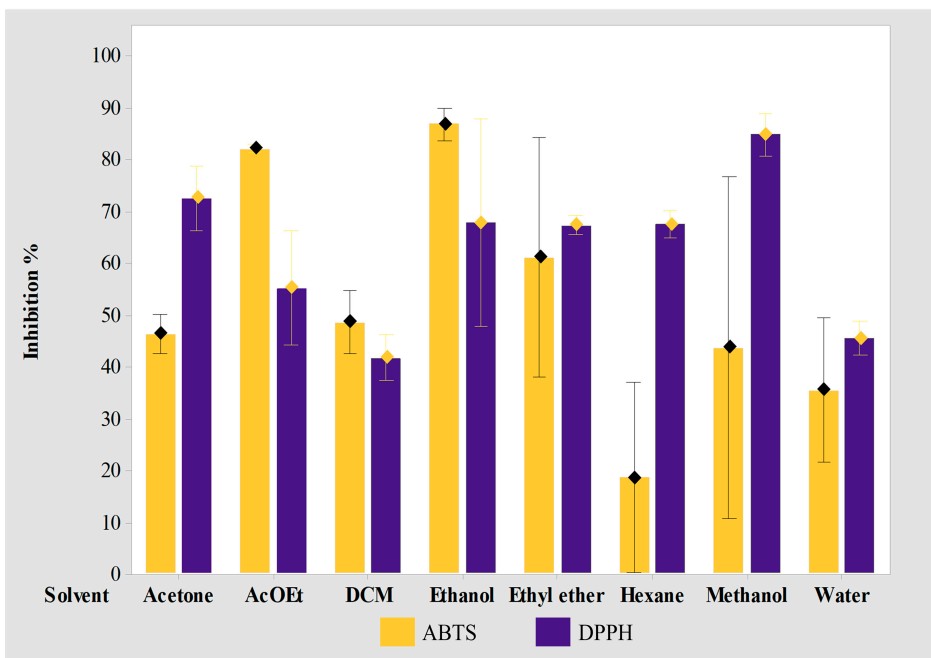

**Figure 8.** Percentage inhibition of *Ginkgo* Biloba extracts, measured by the DPPH and ABTS test.

This indicates that the extract with water or the intake of GB in a traditional infusion is not the most appropriate for an antioxidant effect as shown in the literature. Instead, an extract with ethanol or ethyl acetate may contribute best to a more powerful antioxidant effect, which implies the need for a viable formulation for its administration as capsules or tablets.

## 4. Conclusions

*Gingko biloba* extracts have shown benefits in patients with Alzheimer's disease; however, there are not many studies on the mechanism. With bioinformatic tools from the metabolites reported for GB, it has been shown that they can affect the CNS through different pathways: the monoamine oxidase pathway (isoform A and B), the cholinergic pathway (acetylcholinesterase and butylcholinesterase), and the APP pathway (β-secretase and γ-secretase), proposing an explanation for the effects reported at the molecular level. This is in addition to a second approach through the inhibition of ROS, demonstrating that the polar extracts of GB have an inhibition percentage greater than 50%, resulting in a dual potential against AD via antioxidants and neurotransmitters and via amyloid plaque formation.

**Author Contributions:** Conceptualization, A.C.-C. and J.F.-H.; methodology, A.C.-C. and J.S.-R.; software, A.C.-C. and E.S.-G.; validation, A.C.-C. and J.F.-H.; formal analysis, E.S.-G.; investigation, E.S.-G. and J.S.-R.; resources, J.F.-H.; data curation, A.C.-C. and J.S.-R.; writing—original draft preparation, E.S.-G. and A.C.-C.; writing—review and editing, J.S.-R. and J.F.-H.; project administration, J.F.-H. and A.C.-C.; funding acquisition, J.F.-H. All authors have read and agreed to the published version of the manuscript.

**Funding:** This research was funded by CONACYT (grant PRONACES-31758), and the APC was funded by Instituto de Fisiología-VIEP-BUAP.

**Institutional Review Board Statement:** Not applicable.

**Informed Consent Statement:** Not applicable.

**Data Availability Statement:** All data are available in the text.

**Acknowledgments:** Allen Coombes for language review.

**Conflicts of Interest:** The authors declare no conflict of interest.

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
