# Peer review of "Ginkgo biloba: Antioxidant Activity and In Silico Central Nervous System Potential"

_cimb, doi:10.3390/cimb45120604_

Round 1

Reviewer 1 Report

The manuscript discusses the importance of Ginkgo biloba and its chemical constituents on CNS through an in-silico approach.

 There are a few concerns that need to be addressed.

1. According to ADME studies, Most of the compounds are polar and do not cross BBB, so these are considered potential drug candidates for CNS.

2. The antioxidant studies do not provide any evidence of chemical constituents. Also, the study does not relate to the rest of the manuscript. What authors want to get from this study. Similar antioxidant studies were reported in the literature as well.

3. The conclusion part is confusing and does not summarize the study adequately. 

4. The flavonoids discussed in this manuscript are also present in other plants, so how can authors claim that GB extracts for AD?

Author Response

Dear Reviewer, we appreciate your comments and corrections. The corrections made are attached.

1. According to ADME studies, Most of the compounds are polar and do not cross BBB, so these are considered potential drug candidates for CNS.

It has been emphasized in the manuscript which ones cross, as well as transport alternatives reported for this type of metabolites and their effect on CNS, adding support to explain their effect.

2. The antioxidant studies do not provide any evidence of chemical constituents. Also, the study does not relate to the rest of the manuscript. What authors want to get from this study. Similar antioxidant studies were reported in the literature as well.

A more detailed discussion of the antioxidant results has been included, as well as the introduction has emphasized the relevance of the antioxidant capacity with the capture and neutralization of reactive oxygen species, to explain the consistency of this section.

3. The conclusion part is confusing and does not summarize the study adequately.

It has been restructured and focused on the most relevant results of the work.

4. The flavonoids discussed in this manuscript are also present in other plants, so how can authors claim that GB extracts for AD?

It is consistent with the fact that they are found in other plants, and references to their uses on CNS have been added, in addition to the approach being restructured towards the fact that GB extract has already been previously used against CNS and therefore its study becomes of interest. of metabolites reported for proteins associated with AD, as well as the relationship of antioxidant potential against it.

In addition, references have been updated and a more detailed description of the methods used has been included.

Reviewer 2 Report

The manuscript by E. Suárez-Gonzálezet al., entiled Ginkgo biloba: antioxidant activity and in silico central nervous system potential, presents an interesting and vast study on GB extracts mechanisms against Alzheimer Disease.

The study is well-conducted and written, but, some modifications are required:

Please, briefly describe, previously reported methodology an protocols mentioned in sections 2.1. and 2.4.

Figure 8 is not visible.

Please, remove the sentence: So, for practical purposes, the solvent must be evaporated and the residue redissolved to form an aqueous solution, as the aqueous solution is not exactly the most preferable and used form for oral intake of the extract. More likely, it will be ingested under the form of tablets or capsules.

Minor editing is needed

Author Response

Dear Reviewer, we appreciate your comments and corrections. The corrections made are attached.

1. Please, briefly describe, previously reported methodology an protocols mentioned in sections 2.1. and 2.4.

Each of the protocols have been described in more detail and a bibliography has been included for consultation.

2. Figure 8 is not visible.

The quality has been corrected and improved

3. Please, remove the sentence: So, for practical purposes, the solvent must be evaporated and the residue redissolved to form an aqueous solution, as the aqueous solution is not exactly the most preferable and used form for oral intake of the extract. More likely, it will be ingested under the form of tablets or capsules.

The statement has been removed and adjusted to the purpose of the manuscript.

In addition, references have been updated and included a more detailed description of the methods used and language revision throughout the manuscript.

Round 2

Reviewer 1 Report

The authors have updated the manuscript, and I think it is ready for acceptance.

Reviewer 2 Report

Thank you for your comments and adjustment of the manuscript.